# Deep Learning-Based Real-Time Multiple-Object Detection and Tracking from Aerial Imagery via a Flying Robot with GPU-Based Embedded Devices

**DOI:** 10.3390/s19153371

**Published:** 2019-07-31

**Authors:** Sabir Hossain, Deok-jin Lee

**Affiliations:** School of Mechanical & Convergence System Engineering, Kunsan National University, 558 Daehak-ro, Gunsan 54150, Korea

**Keywords:** multi-target detection and tracking, multi-copter drone, aerial imagery, image sensor, deep learning, GPU-based embedded module, neural computing stick, image processing

## Abstract

In recent years, demand has been increasing for target detection and tracking from aerial imagery via drones using onboard powered sensors and devices. We propose a very effective method for this application based on a deep learning framework. A state-of-the-art embedded hardware system empowers small flying robots to carry out the real-time onboard computation necessary for object tracking. Two types of embedded modules were developed: one was designed using a Jetson TX or AGX Xavier, and the other was based on an Intel Neural Compute Stick. These are suitable for real-time onboard computing power on small flying drones with limited space. A comparative analysis of current state-of-the-art deep learning-based multi-object detection algorithms was carried out utilizing the designated GPU-based embedded computing modules to obtain detailed metric data about frame rates, as well as the computation power. We also introduce an effective target tracking approach for moving objects. The algorithm for tracking moving objects is based on the extension of simple online and real-time tracking. It was developed by integrating a deep learning-based association metric approach with simple online and real-time tracking (Deep SORT), which uses a hypothesis tracking methodology with Kalman filtering and a deep learning-based association metric. In addition, a guidance system that tracks the target position using a GPU-based algorithm is introduced. Finally, we demonstrate the effectiveness of the proposed algorithms by real-time experiments with a small multi-rotor drone.

## 1. Introduction

Target detection has attracted significant attention for autonomous aerial vehicles due to its notable benefits and recent progress. Target tracking with an unmanned aerial vehicle (UAV) can be used for intelligence, surveillance, and reconnaissance missions [1]. Target tracking can be used in autonomous vehicles for the development of guidance systems [2]. Pedestrian detection [3], dynamic vehicle detection, and obstacle detection [4] can improve the features of the guiding assistance system. Object recognition technologies for self-driving vehicles have strict requirements in terms of accuracy, unambiguousness, robustness, space demand, and costs [5]. Similarly, object recognition and tracking features in an aerial vehicle can assist in drone navigation and obstacle avoidance. Visual recognition systems in a UAV can be used in many applications, like video surveillance, self-driving systems [6], a panoramic aerial view for traffic management, traffic surveillance, road conditions, and emergency response, which has been the interest of transportation departments for many years [2,7].

Previously, target detection in drone systems mostly used vision-based target finding algorithms. For example, a Raspberry Pi and OpenCV were used to find a target [8]. However, computer vision techniques might provide less accurate results and have issues in predicting unknown future data. On the other hand, machine learning target-detection algorithms can provide a very accurate result, and the model can make predictions from unknown future data. Visual recognition systems involving image classification, localization, and segmentation have accomplished extraordinary research contributions [6]. Moreover, deep learning has made great progress in solving issues in the fields of computer vision, image and video processing, and multimedia [9]. Because of the critical advancements in neural networks, particularly deep learning [10], these visual recognition systems have shown great potential in target tracking.

On-board and off-board ground-based systems are promising platforms in this context. Most of the time, the aerial vehicle system cannot be equipped with heavy devices due to weight and power consumption. Therefore, off-board ground systems play a vital role. In some cases, communication with the ground station could be impossible due to distance or coverage. An on-board system that can support both weight and power consumption would be a perfect framework for such a situation and environment.

An embedded real-object detection system was developed for a warning system using a UAV [11], but they used only one specific algorithm with different resolutions as an input and one specific embedded module. In the present study, we used various algorithms and a different embedded system to execute the algorithms. The deep drone project used a Jetson GPU and faster R-CNN to detect and track objects [12]. In our study, we used Jetson AGX Xavier for better performance.

A GPU enhances the performance in a deep learning-based visual recognition system. However, such a system also has some disadvantages like more power consumption than CPUs, and they are obviously costlier than an embedded CPU system. These situations can be overcome by implementing a neural computing stick with a CPU device, which is a constraint for executing deep learning models. However, the developed system can be forced to perform efficiently by optimization in the processing unit. In this paper, we discuss an on-board and off-board system that was developed for an aerial vehicle using well-known object detection algorithms.

## 2. Hardware Development of the Drone Framework

The target detection and tracking system can be easily implemented in an aerial vehicle. The structure of the aerial vehicle is the same for all the algorithms except for some minor changes in the embedded system and the exterior body structure. Figure 1 presents the 3D CAD design of our aerial vehicle. Table 1 presents the primary physical specifications of the drone system and the camera mounted on the drone. Figure 2 illustrates the embedded hardware setups used for the target detection and tracking system. Figure 3 shows the detailed hardware setup of the embedded systems used for the detection and target tracking system.

### 2.1. Technical Specifications of Different Embedded Devices Used for Target Detection and Tracking

Our main focus was on Jetson modules [13] for target detection. Other than the Jetson modules, we used GPU-constrained devices like Raspberry Pi [14], Latte Panda [15], and Odroid Xu4 [16]. A Movidius NCS [17] was used to boost the processing power of these limited devices. Moreover, we tried a different approach where we transmitted the aerial image data to the ground station that was equipped with GTX 1080 [18]. The target detection output was directly from the ground station. Below, we give a brief discussion about the embedded devices.

#### 2.1.1. NVidia Jetson Modules (TX1, TX2, and AGX Xavier)

NVidia Jetson devices are embedded AI computing platforms that provide high-performance, low-power computing support for deep learning and computer vision. Jetson modules can be flashed with NVidia JetPack SDK, which contains TensorRT, OpenCV, CUDA Toolkit, cuDNN, and L4T with the LTS Linux Kernel [19].

Jetson TX1 is the world’s first supercomputer on a module and can provide support for visual computing applications. It is built with the NVidia Maxwell™ architecture and 256 CUDA cores delivering performance of over one teraflop [20].

Jetson TX2 is one of the fastest, most power-efficient embedded AI computing devices. This 7.5-watt supercomputer on a module brings true AI computing at the edge. An NVidia Pascal™-family GPU was used to build it and loaded with 8 GB of memory and 59.7 GB/s of memory bandwidth. It included an assortment of standard equipment interfaces that make it simple to incorporate into a wide scope of hardware [21].

Jetson AGX Xavier has exceeded the limit capabilities of previous Jetson modules to a great extent. In terms of performance and efficiency in deep learning and computer vision, it has surpassed the world’s most autonomous machines and advanced robot [22]. This powerful AI computing GPU workstation works under 30 W. It was built around an NVidia Volta™ GPU with Tensor Cores, two NVDLA engines, and an eight-core 64-bit ARM CPU. NVidia Jetson AGX Xavier is the most recent expansion to the Jetson stage [23]. This AI GPU computer can provide unparalleled 32 TeraOPS (TOPS) of the peak computation in a compact 100-mm × 87-mm module form-factor [24]. Xavier’s energy efficient module can be deployed in next-level intelligent machines for end-to-end autonomous capabilities. Table 2 shows the basic comparison between all of them.

#### 2.1.2. GPU-Constraint Devices (Raspberry Pi 3, Latte Panda, and Odroid Xu4)

Raspberry Pi 3 is a tiny, credit card-sized, inexpensive, single-board computer that can be used with a display, mouse, keyboard, power supply, and micro SD card with an installed Linux distribution. It can be used as a fully-fledged computer with basic computer tasks like games, spreadsheet work, etc. Raspberry Pi is used mainly to construct hardware projects, improving programming skill, house automation, and industrial appliances [25,26]. Raspbian OS [27] was used to flash the Raspberry Pi 3.

Latte Panda is the first development board to operate a full version of Windows 10 OS. It is turbocharged with an Intel Quad Core processor and has excellent connectivity, with three USB ports and integrated WiFi and Bluetooth 4.0 [28]. The Arduino co-processor inside Latte Panda can be used to control interactive devices using thousands of plug-and-play peripherals. However, we flashed the whole system with the Lubuntu [29] Linux OS to suit our purpose.

ODROID-XU4 is a very powerful and energy-efficient computing device in a small form factor that offers open-source support with various types of OS, Ubuntu 16.04, Android 4.4 KitKat, 5.0 Lollipop, and 7.1 Nougat. ODROID-XU4 has amazing data transfer speeds because of the support of eMMC 5.0, USB 3.0 and gigabit Ethernet interfaces, which are required to support advanced processing power on ARM devices. Therefore, it provides faster booting, web browsing, networking, and 3D gaming [30,31]. Table 3 shows the basic comparison between all of them.

#### 2.1.3. Movidius Neural Computing Sticks

Deep neural networks (DNNs) can be deployed through Intel Movidius Neural Compute Stick (NCS) using the Intel Movidius Neural Compute SDK (NCSDK) in the constrained devices such as Raspberry Pi, Latte Panda, and Odroid. The Intel Movidius Neural Compute API (NCAPI) is included in the NCSDK to compile, profile, and validate DNN using C/C++ or Python [17]. The NCSDK has two general usages [32]: The tool in NCSDK can be used for profiling, tuning, and compiling a DNN model on the host system.NCAPI can be used to access the neural computing device hardware to accelerate DNN inferences by prototyping a user application on the host system.

The NCS was designed for image processing using Deep Learning models. Image processing is very resource intensive and often runs slowly on devices such as the Raspberry Pi, Latte Panda, and Odroid. The Movidius NCS speeds up the deep learning-based model on constrained devices that have less processing power for deep learning models. The constrained devices are Raspberry Pi, Latte Panda, Odroid, etc. The dimension of the NCS is about 7 cm × 3 cm × 1.5 cm, and it has a USB3 Type A connector. NCS has a low power high-performance visual processing unit (VPU) similar to the “follow me” mode of DJI drones for visual-based functionality [33]. It can save space, money, bandwidth, weight, and power while building drone hardware.

### 2.2. The Architecture of the Developed Embedded System

In this section, we discuss elaborately the systems that we used with the main framework of the drone. The purpose of the vehicular system that we developed in this study was to find targets while navigating and following paths, as shown in Figure 4. Initial maneuver is take-off from the ground, as shown in the 1st and 2nd position of the drone and look for target, as shown in the 3rd position of the drone in Figure 4. Moreover, our vision was to compare all the systems with respect to a different point of view.

#### 2.2.1. On-Board GPU System

The on-board GPU system takes the aerial footage and feeds the data to the onboard TX1/TX2, as shown in Figure 5a, b. The system generates a detection result frame with a name and confidence percentage. In this process, TX1/TX2 is responsible for processing the whole algorithm and streaming the data over a network. The ground station that is connected with a similar network receives the streaming data and displays the results. From the monitoring output, it is also possible to obtain the detected target name, confidence, and notification. Later, we implemented similar steps for the Jetson AGX Xavier mentioned, as shown in Figure 5c. It seems that the powerful Xavier GPU system can perform well and efficiently with object detection algorithms like YOLO, SSD, and R-CNN.

#### 2.2.2. Off-Board GPU-Based Ground Station

An on-board system usually performs all the work by itself, but in this system, the on-board device performed only part of the job. The on-board device was responsible for streaming the captured image data. A real-time imaging processor and transmission system setup were established on the UAV to communicate with the ground station. The GPU-based ground station received the data, and the algorithm used the GTX 1080 GPU system [18] to process the raw image data.

Figure 6 shows a diagram of the whole off-board GPU-based ground station. Later, the detected target is displayed with the name and confidence on the monitor of the ground workstation. We used Python socket [34] to transmit the data over the network to the specific IP address of the drone. The GTX 1080 is a very powerful system for detecting an object or target and is fast and smooth in this system.

#### 2.2.3. On-Board GPU-Constrained System

To construct the on-board system, we used inexpensive devices like the Raspberry Pi, which we combined with a neural stick. We tried SSD(Single Shot Multibox Detector)-Mobilenet [35,36] and the YOLO object detection algorithm [37] in this framework. Similarly, we tried Latte Panda and Odroid XU4 as the replacement for the Raspberry Pi to check the performance of the output results, as shown in Figure 7. We also performed an experiment with SSD-Mobilenet without the use of the neural stick in the system.

### 2.3. Python Socket Server to Send an Image to the GPU-Based Ground Station

Socket programming [34] is helpful for communication between a server and a client that are on two different systems. It is a way of connecting two nodes where one node uses a particular IP address to reach another node. A server that will broadcast the aerial image must specify the IP and port using a specific network where the client is also connected. The server initiates and always listens to the incoming connection. On the other hand, the client on the same network reaches out to the server and obtains the broadcast message, which was the aerial image in our case.

Using a GPU-based ground station as a client, we used a continuous image stream for our purposes. This socket program shows a small lag in the incoming image stream depending on the type of network to which both the server and the client are connected. However, it did not drastically affect the performance of target detection. This communication tool was used in our off-board GPU-based ground station system.

## 3. Implemented Object Detection Algorithm in the Drone System

Figure 8 shows a diagram of the deep learning algorithm list used on the embedded systems for target tracking system.

### 3.1. You Only Look Once: Real-Time Object Detection

You only look once (YOLO) [37] is a fast object detection algorithm. Although it is no longer the most accurate object detection algorithm, it is a very good choice when real-time detection is needed without loss of too much accuracy. YOLO uses a single CNN network for both classification and localizing an object using bounding boxes [38]. The architecture of YOLO is shown in Figure 9.

#### 3.1.1. YOLOv2

YOLO provides real-time processing with high accuracy, but it has higher localization errors and lower recall response than other region-based detector algorithms [39]. YOLOv2 [40] is an upgraded version of YOLO that overcomes the lower recall response and increases the accuracy with fast detection. The changes in YOLOv2 are briefly discussed below:The fully-connected layers that are responsible for predicting the boundary box were removed.One pooling layer was removed to make the spatial output of the network be 13 × 13 instead of 7 × 7.The class prediction was moved from the cell level to the boundary box level. Now, each prediction had four parameters for the boundary box [39].The input image size was changed from 448 × 448 to 416 × 416. This created odd-numbered spatial dimensions (7 × 7 vs. 8 × 8 grid cell). The center of a picture is often occupied by a large object. With an odd number of grid cells, it is more certain where the object belongs [39].The last convolution layer was replaced with three 3 × 3 convolutional layers, each outputting 1024 output channels to generate predictions with dimensions of 7 × 7 × 125. Then, a final 1 × 1 convolutional layer was applied to convert the 7 × 7 × 1024 output to 7 × 7 × 125 [39].

#### 3.1.2. YOLOv3

The output object classes were mutually exclusive since classifiers assumed that output labels were mutually exclusive. YOLO had a softmax function to convert scores into probabilities that added up to one. YOLOv3 [41] uses a multi-label classification. Non-exclusive output labels can show a score that is more than one. Instead of using the softmax function, YOLOv3 uses independent logistic classifiers to calculate the likeliness of the input belonging to a specific label. YOLOv3 uses binary cross-entropy loss for each label instead of the mean squared error in calculating the classification loss. Avoiding the softmax function reduces the computation complexity [39]. Figure 10 [42] shows the neural architecture of YOLOv3.

#### 3.1.3. YOLOv2 Tiny and YOLOv3 Tiny

Tiny YOLO is based on the Darknet reference network [43] and is much faster, but less accurate than the normal YOLO model [40,41]. The full YOLOv2 model uses three-times as many layers as tiny and has a slightly more complex shape. The “tiny” version of YOLO has only nine convolutional layers and six pooling layers. Since YOLO tiny uses fewer layers, it is faster than YOLO, but also a little less accurate.

#### 3.1.4. YOLO-9000

YOLO-9000 [40] is a better, faster, and stronger version of YOLO. Below, brief points are shown regarding the matter of making it better, faster, and stronger [44].

The approaches for better are:Batch normalization: Batch normalization was used in all convolutional layers, which helped to obtain more than 2% improvement in mAP (mean average precision).High-resolution classifier: The classification network was fine-tuned on 448 × 448 images instead of trained with 224 × 224 images. This helped the network perform better at higher resolution. This high-resolution classification network gave an increase of almost 4% mAP.Convolutional with anchor boxes: In YOLOv2, anchor boxes are adopted while removing all fully-connected layer. One pooling layer was removed to increase the resolution of the image output. This enabled more boxes to be generated, which improved the recall from 81% (69.5 mAP) to 88% (69.2 mAP) [40].Direct location prediction: Prediction becomes easier if the location is constrained or limited. YOLO9000 predicts location coordinates relative to the location of the grid cell, which bounds the ground truth to fall between zero and one. It does not make predictions by using the offset to the center of the bounding box [44].Fine-grained features: A pass-through layer was included like ResNet to use fine-grained features for localizing smaller objects.Multi-scale training: The same network can predict at different resolutions if a dataset with different resolutions is utilized while training the network. That means that the network can make predictions from a variety of input dimensions.

The approaches for faster are:Instead of using VGG-16, a custom network of 19 convolutional layers and five max-pooling layers was used. The custom network that used by the YOLO framework is called Darnet-19 [45].

The approaches for stronger are:Hierarchical classification: To build a hierarchical prediction, several nodes were inserted. A semantic category was defined for each node at a level. Therefore, different objects in one image can be amalgamated into one label since they were from one higher level semantic label.Joint classification and detection: For training a large-scale detector, two types of datasets were used. a traditional classification dataset that contained a large number of categories and a detection dataset [45].

### 3.2. SSD: Single-Shot MultiBox Detector

A typical CNN network gradually reduces the feature map size and expands the depth toward the deeper layers, as shown in Figure 11. Larger receptive fields are covered by the deep layers, which creates more abstract representation. Smaller receptive fields are covered by the shallow layers. Thus, the network can use this information to predict big objects using deeper layers and to predict small objects using shallow layers [36,46]. The main idea is to use a single network for speed and to remove region proposal. It adjusts the bounding box according to the prediction. The last few layers are responsible for smaller bounding box prediction, which is also responsible for the prediction of different bounding boxes. The final prediction is a combination of all these predictions. To better understand SSD, its structure is explained by its name [47]: Single shot: The tasks of object localization and classification are executed in a single forward pass of the network.MultiBox: MultiBox is the name of a technique for bounding box regression developed by Christian Szegedy et al. for fast class-agnostic bounding box coordinate proposals [48,49].Detector: The classification of a detected object is performed by the network, which is called the object detector.

### 3.3. Region-Based Convolutional Network Method for Object Detection

The goal of R-CNN [50] is to identify the main objects correctly through the bounding box in the image. R-CNN creates bounding boxes of proposed regions using a technique called selective search [51]. At a high level, selective search, shown in Figure 12, looks at the image through boxes of different sizes and each size group together to identify objects. Once the process is completed, R-CNN deforms the region to a standard square size and passes it to the modified version of AlexNet to find the valid region. In the final layer of the CNN, R-CNN adds a support vector machine (SVM), which simply classifies by determining the possibility of finding an object and the object name [52].

#### 3.3.1. Faster R-CNN

R-CNN works really well, but is quite slow. One reason is that it requires a forward pass of the CNN (AlexNet) for every single proposed region for every single image. Another reason is that it has to train three different models separately: the CNN to generate image features, the classifier to predicts the class, and the regression model to tighten the bounding boxes [52]. Faster R-CNN adds a fully-convolutional network on top of the features of the CNN known as the region proposal network to speed up the region proposal [53].

#### 3.3.2. Mask R-CNN

Mask R-CNN is an extended version of faster R-CNN for pixel level segmentation. Mask R-CNN [54] works by inserting a branch into faster R-CNN that adds a binary mask to determine whether a given pixel is part of an object. The branch is a fully-convolutional network on top of a CNN-based feature map [52]. Once these masks are generated, mask R-CNN amalgamates them with the classifications and bounding boxes that result from faster R-CNN. Overall, it generates precise segmentation.

### 3.4. DeepLab-v3 Semantic Segmentation

We implemented Tensorflow’s Deeblap model for real-time semantic segmentation in embedded platforms [55,56]. It is built on top of a powerful convolutional neural network (CNN) backbone architecture [57,58] to obtain the most accurate results, which are intended for server-side deployment. DeepLab-v3 was extended by including a simple decoder module to refine the segmentation results, especially along object boundaries, which are very effective. Depth-wise separable convolution was later used in both decoder modules and atrous spatial pyramid pooling [59]. As a result, the output was a faster and stronger encoder-decoder network for semantic segmentation.

## 4. Implemented Target Tracking Algorithm in the Drone System

There are many target tracking algorithms available for both multiple objects and a single object. Most of them use conventional methods and neural networks for tracking. We implemented Deep SORT [60], which seems feasible to track objects in real-time using our own hardware system.

### 4.1. Deep Sort Using YOLOv3.

The purpose of this project is to add object tracking to YOLOv3 [41] and achieve real-time object tracking using a simple online and real-time tracking (SORT) algorithm with a deep association metric (Deep SORT) [60]. The algorithm integrates appearance information to enhance the efficiency of SORT [61]. Thus, it is possible to track objects for a longer time through visual occlusions. It also effectively reduced the number of identity switches by 45%. A large-scale person re-identification dataset was used in pre-training. According to their paper, experimentation also showed overall performance at high frame rates.

SORT has a flaw in tracking through occlusions since an object typically needs to stay in the frontal view of the camera. This issue is successfully overcome by using a more informed metric that combines motion and appearance information instead of the association metric. Specifically, a convolutional neural network (CNN) was implemented after it was trained using a large-scale person re-identification dataset to distinguish pedestrians.

The Hungarian algorithm was used to solve the association between the predicted Kalman states and newly-arrived measurements. The mathematical formulation was solved by integrating motion and appearance information through the combination of two appropriate metrics. Both metrics were combined using a weighted sum to build the association problem, and the following equation was presented [60].
(1)$$c_{i,j}=\lambda d^{\left(1\right)}\left(i,j\right)+\left(1-\lambda \right)d^{\left(2\right)}\left(i,j\right)$$
where an association admissible is called if it is within the gating region of both metrics [60]:(2)$$b_{i,j}=\overset{2}{\underset{m=1}{\prod }}b_{i,j}^{\left(m\right)}$$

Algorithm 1 was directly taken from Deep SORT [60] and displays the matching cascade algorithm that assigns priority to more frequently-seen objects. The inputs were the track index T, detection index D, and maximum age $A_{max}$. The association cost and gate matrix are computed on Line 1 and Line 2. The linear assignment problem of increasing age was solved by iterating track age *n*. The subset of tracks $T_{n}$ that have not been associated in the last *n* frames of detection is selected in Line 6. The linear assignment between tracks in $T_{n}$ and unmatched detections U is solved in Line 7. The set of matches and unmatched detections is updated in Lines 8 and 9. Later, the set value is returned in Line 11. Like the original SORT algorithm [61], intersection over union association is executed in the last matching stage. The execution of the set of unconfirmed and unmatched tracks of age *n =* 1 helps to account for sudden appearance changes.

**Algorithm 1.** Matching cascade.**Input:** Track indices T={1,…,N}, detection indices D={1,…,M}, maximum age $A_{max}$1:Compute cost matrix $C=\left[c_{i,j}\right]$ using Equation (1)2:Compute gate matrix $B=\left[b_{i,j}\right]$ using Equation (2)3:Initialize the set of matches ℳ←∅4:Initialize the set of unmatched detections U←D5:
**for**
$n\in \left\{1,\;\ldots ,\;A_{max}\right\}$
**do**
6: Select tracks by age $T_{n}\leftarrow \{i\in T\;|\;a_{i}=n\}$7: $\left[x_{i,j}\right]$ ← min cost matching $C,\;T_{n},\;U$8: $ℳ\leftarrow ℳ\;\cup^{\;}\;\left\{\left(i,j\right)\;|\;b_{i,j}.\;x_{i,j}>0\right\}$9: $U\leftarrow U\;\setminus \;\left\{j\;|\;\Sigma_{i}\;b_{i,j}.\;x_{i,j}>0\right\}$10:
**end for**
11:
**return**
ℳ,U


A huge amount of data is needed for feasible people tacking based on deep metric learning. A CNN architecture was trained on a large-scale person re-identification dataset [62], which contains over 1,100,000 images of 1261 pedestrians. Table 4 presents the CNN architecture of its network [60]. A wide residual network [63] with two convolutional layers followed by six residual blocks was used in their architecture. The final batch and $l_{2}$ normalization projected the feature onto the unit hypersphere.

### 4.2. Guiding the UAV toward the Target Using YOLOv2

The purpose of this algorithm was to fly the drone toward the target using only a detection algorithm. In this case, we used a person as a classifier and calculated the area of the bounding box of the person from a certain safe distance, which will be the final goal for the drone. Algorithm 2 presents the guidance algorithm toward the target. We used YOLOv2, but it can be applicable to our algorithms, as well. Using the YOLOv2 algorithm, we easily could obtain coordinates for the bounding box and calculate its center.

Initially, the algorithm either looks for target N=1 in Step 1 or it does nothing and waits for the target like in Step 11. If there is a target and it is a person L=1, the algorithm will loop from Step 4 to Step 8. In Step 4, the area of the bounding box A is calculated as A=height×weight. In Step 5 and Step 6, it calculates $Error_{Center}$ and $Error_{Area}$, the values of which will be used for yaw angle and forward velocity, respectively. The algorithm will perform Step 7 until $Error_{Center}=0$ and Step 8 until $A_{max}\leq A$.

**Algorithm 2.** Following target.**Input:** Class for person L=1, area of the bounding box A=height×weight, maximum area $A_{max}$ = constant, center of the whole image $C_{image}=\mathrm{constast}$**Output:**$Error_{Center}=C_{image}-C_{bounding}$, $Error_{Area}=A_{max}-A$1:Look for target N2:
**if**
N=1
**then**
3: **if**
L=1
**then**4:  Calculate A and $C_{bounding}$5:  $Error_{Center}=C_{image}-C_{bounding}$6:  $Error_{Area}=A_{max}-A$7:  Repeat Step 5 until $Error_{Center}=0$, and use $Error_{Center}$ for yaw angle8:  Repeat Step 6 until $A_{max}\leq A$, and use $Error_{Area}$ for forwarding velocity9: **end if**10:
**else**
11: Do nothing12:
**end if**


## 5. Results

In this section, we present the experimental results from the aerial vehicle and the performance according to the systems we used in the aerial vehicle.

### 5.1. Detection Results with Classification from the Drone Using the On-Board GPU System

Figure 13 below shows the result of target detection with bounding boxes and confidences levels from the on-board GPU system. Figure 13a shows the segmented result from the UAV using Xavier as the on-board GPU system. When the person was small or out of the field of view, this algorithm was unable to detect that person well. On the other hand, using faster R-CNN for detection showed a very accurate result, but revealed a very low performance rate, as shown in Figure 13b,d. The YOLOv2 algorithm revealed a satisfying result in terms of performance rate, but it could only detect the person from less than 20 m away, as shown in Figure 13c,g. However, it is possible to increase the input image dimension in order to detect a person from a far distance. In that case, the performance rate will be reduced in terms of frames per second (FPS). Our input dimensions were 416×416 for all the YOLO detection algorithms. More significantly, YOLOv3 showed more accurate results even from a far distance because of its powerful 75 convolutional layers, as shown in Figure 13e,f,i**.**

### 5.2. Detection Results with Classification from the Drone Using the GPU-Based Ground System

Figure 14 shows the target detection results of the GPU-based ground system. Figure 14a shows the segmented result from the off-board GPU system. The aerial image input came from the Odroid XU4 device, which was attached to the drone. Figure 14b,h shows the output of YOLOv3 in this system. Moreover, we implemented mask R-CNN using this system, as shown in Figure 14d. This system had a latency gap depending on the communication system, but it did not affect the performance rate of the algorithm. A wide-range of WiFi network modules was used to stay connected to the network for both the client and the server.

### 5.3. Performance Results between On-Board and Off-Board System

The performance results in the form of FPS are shown in Table 5. This table provides a quantitative comparison between the on-board embedded GPU system and the off-board GPU-based ground station. This quantitative comparison will change with the input dimension of the image. However, using this table, one can choose the best algorithm and system for a specific operation.

### 5.4. Deep SORT Tracking Result Using Xavier and the Off-Board GPU-Based Ground System

Figure 15 shows the output result of the target tracking algorithm using YOLOv3 in both the Xavier system and the GPU-based ground system.

### 5.5. The Object Detection Result of the On-Board GPU-Constrained System

Figure 16 shows the output result of object detection from Movidius NSC with Odroid xu4. This system is a suitable package for low-payload UAV and a less expensive system. Although the performance rate was not as satisfactory as that of an on-board embedded system, a small task could be done using this system. It is a valid option if either cost, space, or thermal limitation is considered. Movidius has its own heatsink that is made of the metal fin to help with cooling.

### 5.6. Performance Results of the On-Board GPU-Constrained System

The quantitative performance results in the form of FPS are shown in Table 6 for the on-board GPU-constrained systems like Movidius NCS with Raspberry Pi, Latte Panda, and Odroid xu4. This system has the potential to accelerate the interface where a low-power edge device cannot perform it alone.

### 5.7. Monitored Result of Power Consumption

Figure 17 presents the average GPU percentage usage while running different algorithms in the embedded system like AGX Xavier and TX2. All the algorithms used their pre-trained model provided with it.

Figure 18 presents the memory usage in MB and the average CPU percentage usage while running different algorithms in the embedded systems like AGX Xavier. It seems that the CPU percentage was almost similar for all the algorithm (near 4200). In the case of TX2, the CPU usage was near 3500 for all the algorithms.

### 5.8. Guiding the UAV toward the Target

Figure 19 represents the error plot between the center of the image and the center of the target using the YOLOv2 algorithm. The algorithm reduced the error between the target and image center using the yaw movement of the drone. Meanwhile, it also attempted to reduce the error between the area of the bounding box of the target and the maximum given area using forward movement. The forward movement stopped when it reached the maximum given area specified for it. Similarly, if the error increased in a negative direction, the UAV used backward movement.

The TX2 embedded module was used to guide the UAV. The error between two centers was used as feedback to the yaw angle of the vehicle. The final error was equivalent to zero, and the UAV was less than 1 m from the target. We could use this algorithm to follow the target. Accuracy and high frame rate were the major requirements for this algorithm. Due to inconsistent detection, the UAV would cause shaky movement, and a low performance rate would sabotage the process of the algorithm.

## 6. Discussion

Since the main focus of this paper was on target detection from a drone, we first needed to figure out which algorithm provided faster and more accurate results even from far distances. Even though YOLOv2 was faster and accurate, it could not detect objects properly from a far distance. If the distance between the target and drone was more than 20 m, YOLOv2 weight became unable to detect a human. Because of YOLOv3’s architecture, it could detect a target even at 50 m away from the drone. Therefore, we tried to implement Deep SORT with YOLOv3 in a Jetson Xavier for tracking a target.

The YOLO tiny version is not suitable for target detection since it is inaccurate and it is hard to detect objects from a far distance. The target tracking algorithm worked well from 20–30 m away, because within this range, the resolution of the feature from the image remained visible to track the features. It became harder to track properly if the tracking object lost its feature due to far distances. Deep SORT provided a tracking result by counting a target of similar features. However, in some case, it lost track of a counted feature and considered it as a new target.

It is clear that there was an influence of the input resolution on the neural network of the algorithm. The detection frame rate changed with respect to the input dimension that was fed to the neural architecture of the YOLO [11]. In the case of YOLO, we used 416×416 as our input dimension in the configuration file of the YOLO while executing the target detection algorithm. More significantly, the guiding algorithm to follow the target for a person worked less than 20 m away from the target since person detection using YOLOv2 did not work after that limit. The guiding algorithm was purely based on the detection result and the coordinates of the bounding boxes. A poor detection result or poor tracking of the bounding box diverted the drone in the wrong direction. This algorithm also had another limitation. When it faced multiple targets of a similar class, it chose randomly in between them to track. Further research is needed to make the algorithm more robust for such scenarios.

## 7. Conclusions

From the experiments on different GPU systems, it was evident that Jetson AGX Xavier was powerful enough to work as a replacement of a GPU system like NVidia GTX 1080. All sorts of contemporary target detection algorithm performed very well in Jetson Xavier. Jetson TX1 is feasible if the user uses a small weight or model like YOLOv2 tiny. Because YOLOv2 and v3 tiny showed reasonable FPS results for object detection, they were not good enough to detect a target from a far distance. Moreover, the confidence output for using the weight of YOLO tiny was very low.

Jetson Tx2 is a moderate GPU system. The performance was not like that of the Xavier, but it showed outstanding results in the case to YOLOv2 and SSD-Caffe. If there was a limitation in drone weight and power consumption, a neural computing stick attached to the system was quite helpful. Among the three on-board GPU-constrained systems, Odroid XU4 with NCS showed better performances. We also presented the algorithm procedure for tracking with the respective embedded system. We also presented the runtime, GPU consumption, and size of the platform used for the experiment.

## Figures and Tables

**Figure 1 sensors-19-03371-f001:**
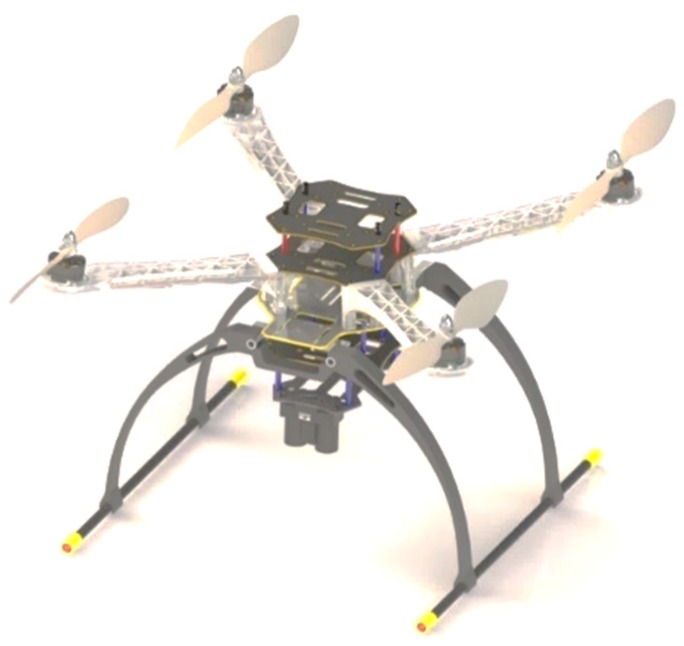
3D CAD design of the external structure of an aerial vehicle with a camera mounted on it.

**Figure 2 sensors-19-03371-f002:**
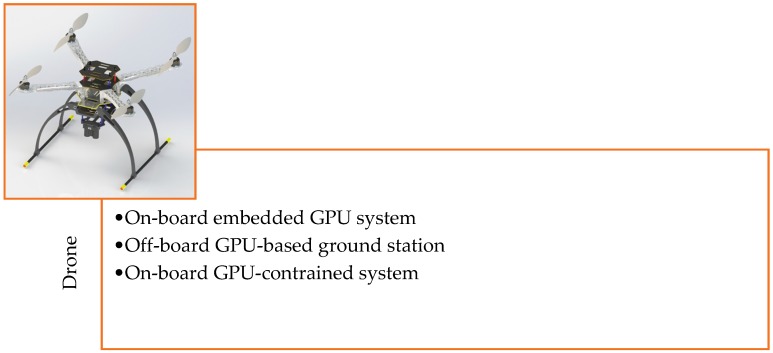
List of developed embedded system types for the aerial vehicle.

**Figure 3 sensors-19-03371-f003:**
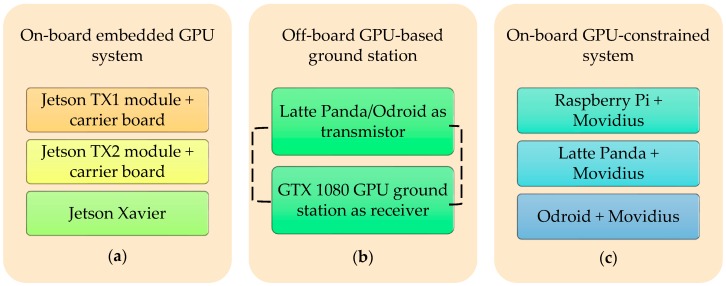
List of the embedded systems implemented inside the common drone structure. (**a**) On-board embedded GPU system; (**b**) off-board GPU-based ground station; (**c**) on-board GPU-constrained system.

**Figure 4 sensors-19-03371-f004:**
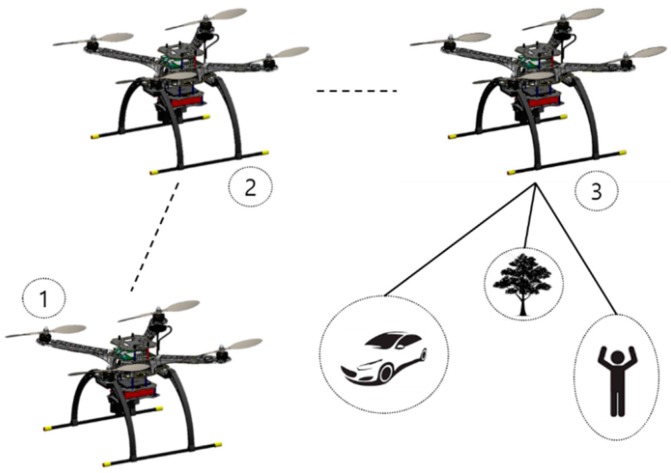
The portrayal of the aerial vehicle looking for a target with a drone taking off from the ground, following a designated waypoint, and looking for objects.

**Figure 5 sensors-19-03371-f005:**
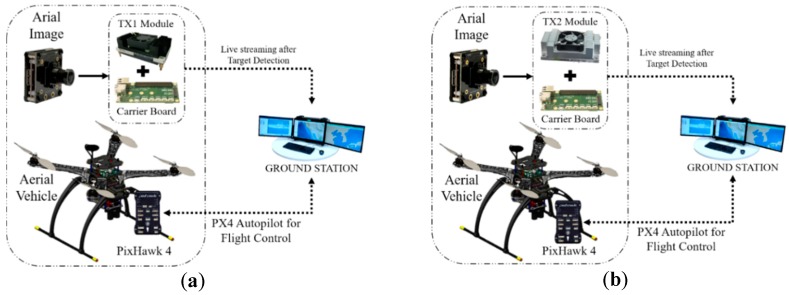
(**a**) On-board target detection system using Jetson TX1, (**b**) On-board target detection system using Jetson TX2, and (**c**) On-board target detection system using Jetson AGX Xavier.

**Figure 6 sensors-19-03371-f006:**
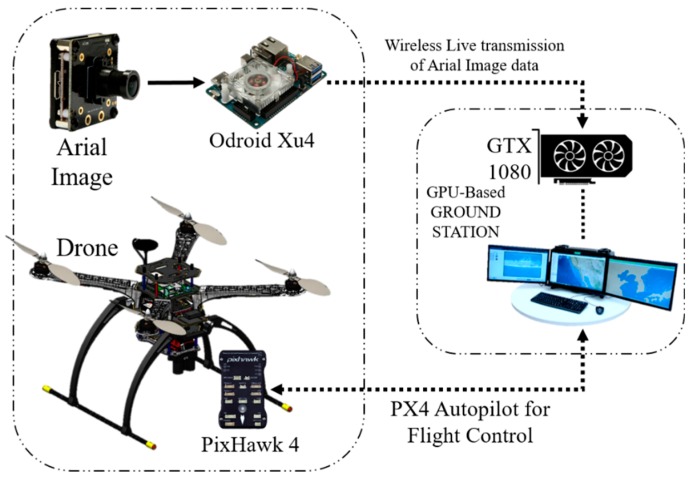
Target detection using the GPU-based ground station from an aerial vehicle.

**Figure 7 sensors-19-03371-f007:**
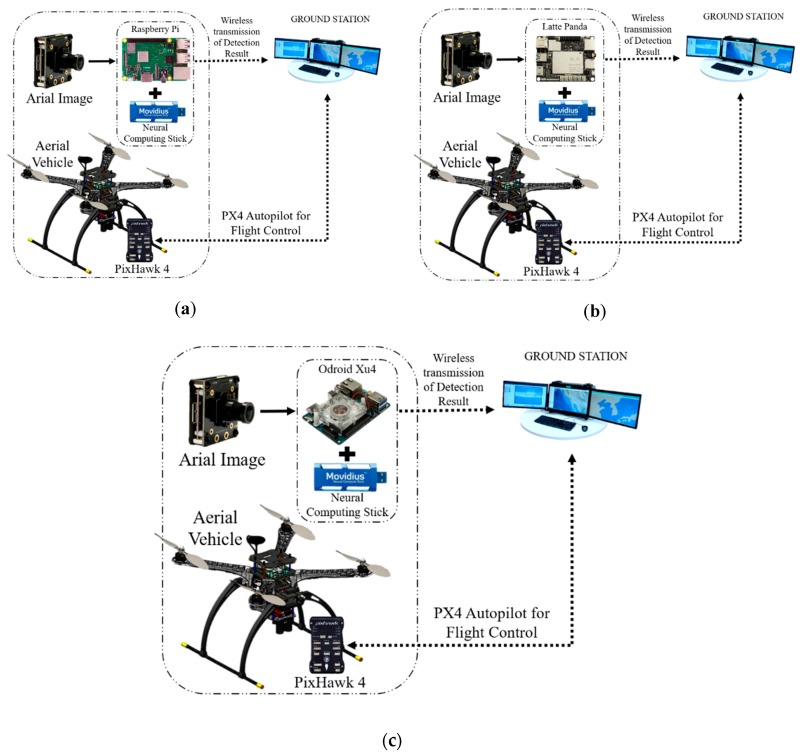
On-board target detection drone system using GPU-constrained devices like (**a**) Raspberry Pi, (**b**) Latte Panda, and (**c**) Odroid Xu4.

**Figure 8 sensors-19-03371-f008:**
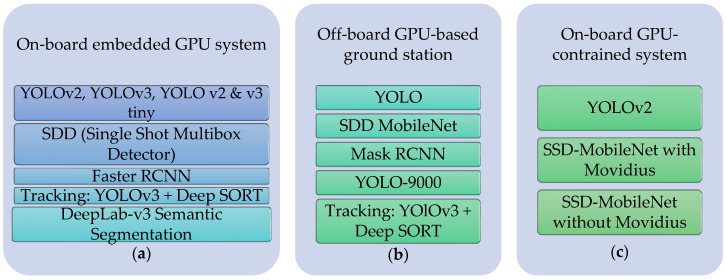
List of deep learning algorithms implemented in the aerial vehicle: (**a**) On-board embedded GPU system; (**b**) off-board GPU-based ground station; (**c**) on-board GPU-constrained system.

**Figure 9 sensors-19-03371-f009:**
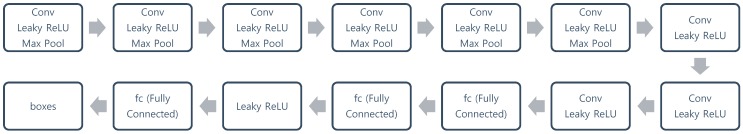
Architecture of YOLO.

**Figure 10 sensors-19-03371-f010:**
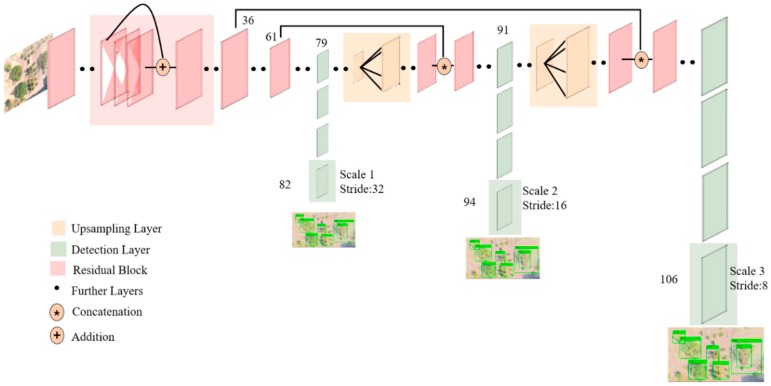
YOLOv3 architecture.

**Figure 11 sensors-19-03371-f011:**
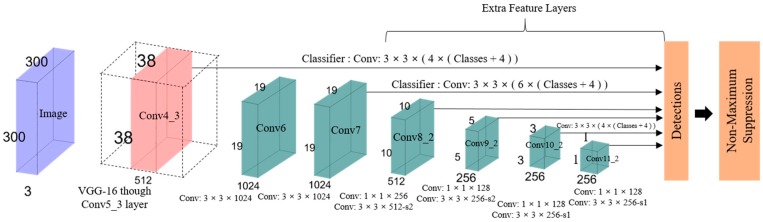
The architecture of SSD.

**Figure 12 sensors-19-03371-f012:**
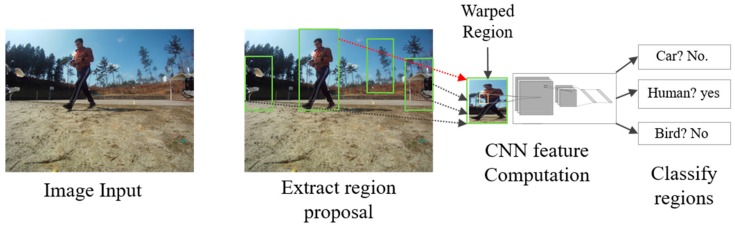
After creating a set of region proposals, R-CNN uses a modified version of AlexNet to determine the valid region.

**Figure 13 sensors-19-03371-f013:**
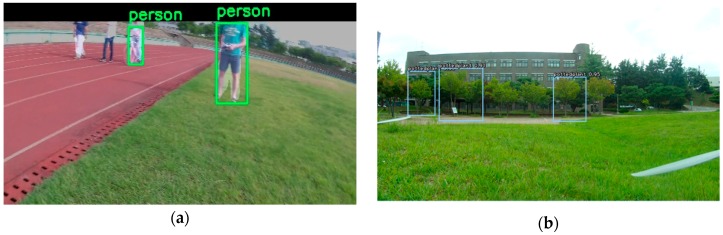
Target detection results of (**a**) real-time segmentation using DeepLabv-3 using Xavier, (**b**) faster R-CNN using TX2, (**c**) YOLOv2 using TX2, (**d**) faster R-CNN using Xavier, (**e**) YOLOv3 using TX2, (**f**) YOLOv3 using Xavier, (**g**) YOLOv2 using Xavier, (**h**) SSD-caffe using Xavier, (**i**) YOLOv3 using Xavier, and (**j**) SSD-caffe using TX2.

**Figure 14 sensors-19-03371-f014:**
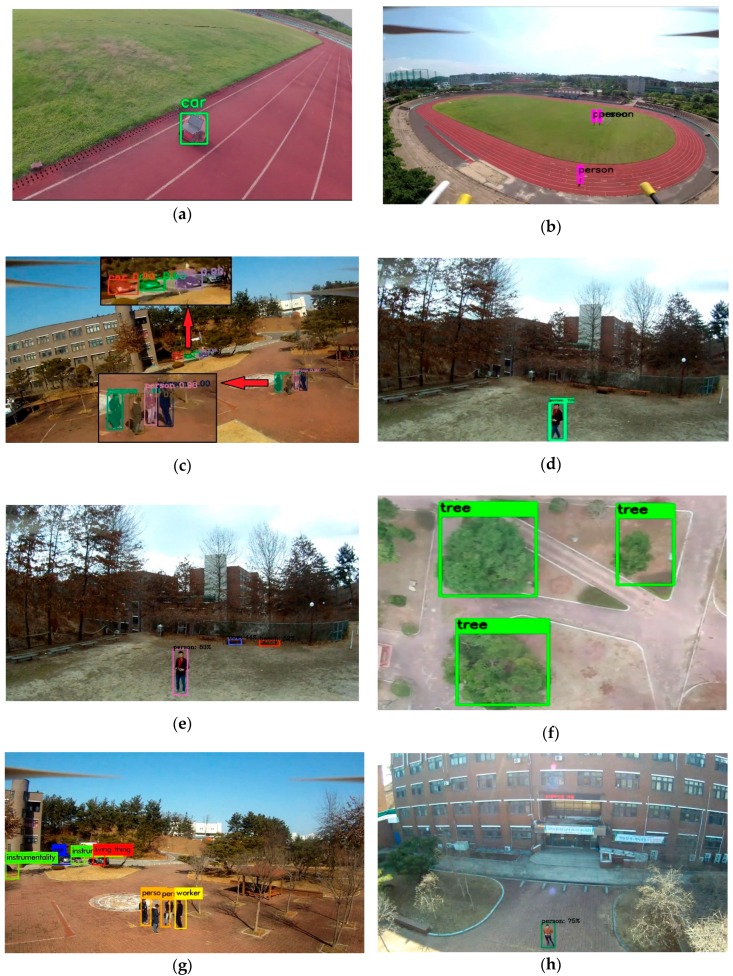
Target detection results using the off-board GPU-based ground station: (**a**) Real-time segmentation using DeepLabv-3, (**b**) YOLOv3, (**c**) mask-R-CNN, (**d**) SSD-Mobilenet, (**e**,**f**) YOLOv2, (**g**) YOLO-9000m, and (**h**) YOLOv3.

**Figure 15 sensors-19-03371-f015:**
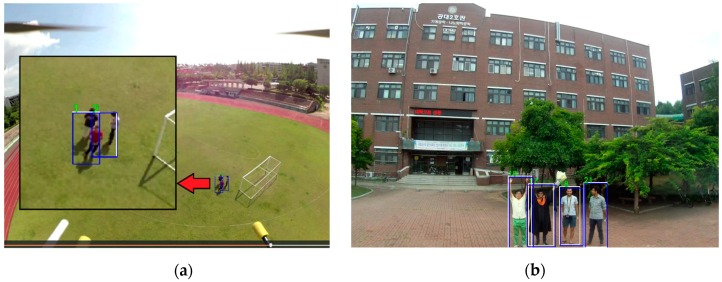
Target tracking results using Deep SORT and YOLOv3 from the drone using (**a**) Xavier and (**b**) the off-board GPU-based ground station.

**Figure 16 sensors-19-03371-f016:**
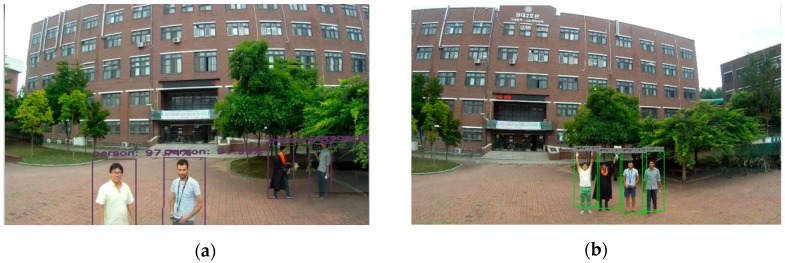
Target detection results from the drone of (**a**) YOLO tiny using Odroid XU4 + NCS and (**b**) SSD-MobileNet using Odroid XU4 + NCS.

**Figure 17 sensors-19-03371-f017:**
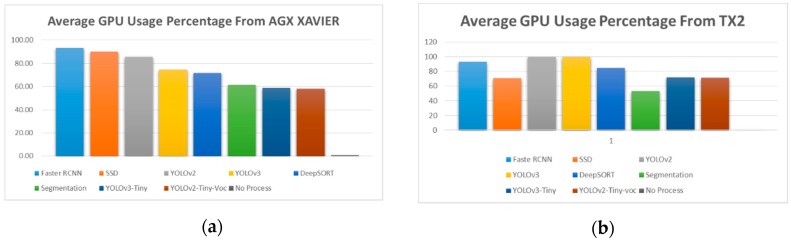
Average GPU usage percentage from (**a**) Xavier and (**b**) TX2.

**Figure 18 sensors-19-03371-f018:**
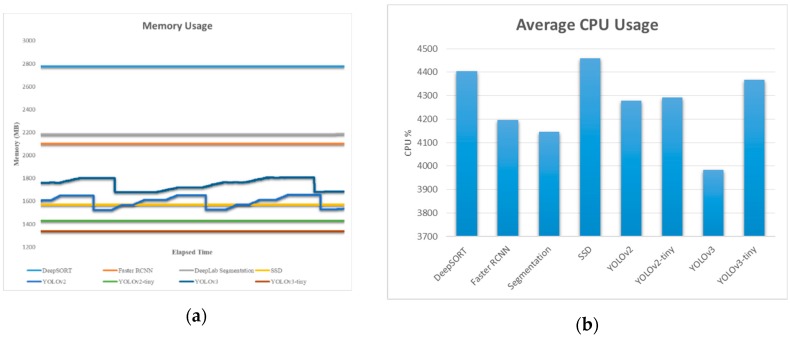
(**a**) Memory usage and (**b**) average CPU usage percentage in Xavier.

**Figure 19 sensors-19-03371-f019:**
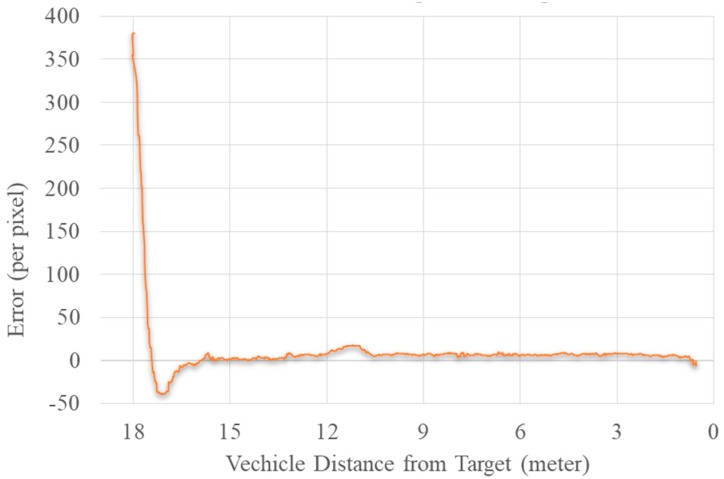
Plot diagram of the error between the image and target center vs. distance.

**Table 1 sensors-19-03371-t001:** Specifications of the drone and camera.

System	Patch Size/Stride	Output Size
UAV	Configuration type	X-configuration
	Dimension (including propeller)	30 cm×30 cm×25 cm(l×w×h)
	Durance	20 min
	Payload (total)	2.5 kg
	Altitude	300 m
Camera (oCam: 5-MP USB 3.0 Camera)	Dimension	42 mm×42 mm×17 mm(l×w×h)
	Weight	35 g
	Resolution	1920 × 1080@30 fps

**Table 2 sensors-19-03371-t002:** Comparison between Jetson modules used for target detection and tracking.

	TX1 ^1^	TX2 ^2^	AGX XAVIER ^3^
GPU	NVidia Maxwell™ GPU with 256 NVidia^®^ CUDA^®^ Cores	NVidia Pascal™ architecture with 256 NVidia CUDA cores	512-core Volta GPU with Tensor Cores
DL Accelerator	None	None	(2x) NVDLA Engine
CPU	Quad-core ARM^®^ Cortex^®^-A57 MPCore Processor	Dual-core Denver 2 64-bit CPU and quad-core ARM A57 complex	8-Core ARM v8.2 64-bit CPU, 8-MB L2 + 4 MB L3
MEMORY	4 GB LPDDR4 Memory	8 GB 128-bit LPDDR4	16 GB 256-bit LPDDR4x | 137 GB/s
STORAGE	16 GB eMMC 5.1 Flash Storage	32 GB eMMC 5.1	32 GB eMMC 5.1
VIDEO ENCODE	4K @ 30	2 × 4K @ 30 (HEVC)	8 × 4K @ 60 (HEVC)
VIDEO DECODE	4K @ 30	2 × 4K @ 30, 12-bit support	12 × 4K @ 30 12-bit support
JetPack Support	Jetpack 2.0~3.3	Jetpack 3.0~3.3	JetPack 4.1.1

^1^ NVidia Jetson TX2 delivers twice the intelligence to the edge. Available online: devblogs.nvidia.com/jetson-tx2-delivers-twice-intelligence-edge (accessed on 3 March 2019). ^2^ JETSON TX2. Available online: www.nvidia.com/en-us/autonomous-machines/embedded-systems/jetson-tx2 (accessed on 3 March 2019). ^3^ Jetson AGX Xavier Developer Kit. Available online: https://developer.nvidia.com/embedded/buy/jetson-agx-xavier-devkit (accessed on 3 March 2019).

**Table 3 sensors-19-03371-t003:** Basic comparison between Raspberry Pi 3, Latte Panda, and Odroid Xu4.

	Raspberry Pi 3 ^1^	Latte Panda ^2^	Odroid Xu4 ^3^
CPU	1.2 GHz 64-bit quad-core ARMv8 CPU	Intel Cherry Trail Z8350 Quad Core 1.44-GHz Boost 1.92 GHz	Samsung Exynos5422 Cortex™-A15 2 GHz and Cortex™-A7 Octa core CPUs
GPU	Broadcom video core 4	Intel HD Graphics, 12 EUs @ 200–500 MHz	Mali-T628 MP6
MEMORY	1 GB	4 GB DDRL3L	2-Gbyte LPDDR3 RAM PoP stacked
STORAGE	Support MicroSD	64 GB eMMC	Supports eMMC5.0 HS400 and/or micro SD

^1^ Raspberry Pi 3 Model B. Available online: www.raspberrypi.org/products/raspberry-pi-3-model-b/ (accessed on 3 March 2019). ^2^ Latte Panda 4G/64G. Available online: www.lattepanda.com/products/3.html (accessed on 3 March 2019). ^3^ Description: Odroid Xu4. Available online: www.hardkernel.com/shop/odroid-xu4-special-price (accessed on 3 March 2019).

**Table 4 sensors-19-03371-t004:** Overview of the CNN architecture.

Name	Patch Size/Stride	Output Size
Conv 1	3×3∕1	32×128×64
Conv 2	3×3∕1	32×128×64
Max Pool 3	3×3∕2	32×64×32
Residual 4	3×3∕1	32×64×32
Residual 5	3×3∕1	32×64×32
Residual 6	3×3∕2	64×32×16
Residual 7	3×3∕1	64×32×16
Residual 8	3×3∕2	128×16×8
Residual 9	3×3∕1	128×16×8
Dense 10		128
Batch and $l_{2}$ normalization		128

**Table 5 sensors-19-03371-t005:** Performance comparison between Jetson modules and GTX 1080 used for target detection and tracking.

	TX1	TX2	Xavier AGX	GPU-Based Ground Station: Gtx 1080
YOLOv2	2.9 FPS	7 FPS	26~30 FPS	28 FPS
YOLOv2 tiny voc	6~7 FPS	15~16 FPS	29 FPS	30+ FPS
YOLOv3	---	3 FPS	16~18 FPS	15.6 FPS
YOLOv3 tiny	9~10 FPS	12 FPS	30 FPS	30+ FPS
SSD	8 FPS	11~12 FPS	35~48 FPS	32 FPS
DeepLab-v3 Semantic Segmentation	--	2.2~2.5 FPS	10 FPS	15~16 FPS
Faster R-CNN		0.9 FPS	1.3 FPS	--
Mask R-CNN	--	--	--	2~3 FPS
YOLOv3 + Deep SORT	--	2.20 FPS	10 FPS	13 FPS

**Table 6 sensors-19-03371-t006:** Performance comparison between Raspberry Pi 3, Latte Panda, and Odroid Xu4.

Systems	YOLO	SSD Mobile Net
Movidius NCS + Raspberry Pi	1 FPS	5 FPS
Movidius NCS + Latte Panda	1.8 FPS	5.5~ 5.7 FPS
Movidius NCS + Odroid	2.10 FPS	7~8 FPS
Just Odroid without Movidius NCS	--	1.4 FPS

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
