# Peer review of "Deep Learning-Based Real-Time Multiple-Object Detection and Tracking from Aerial Imagery via a Flying Robot with GPU-Based Embedded Devices"

_sensors, 2019, doi:10.3390/s19153371_

Round 1

Reviewer 1 Report

This manuscript documents three types of hardware setup for object tracking using UAV and also compares different object recognition algorithms. While I believe the information provided by this manuscript will be useful for readers who are interested in building an object tracking system with UAV, the manuscript in its current form is inappropriate for publication.

Firstly, the manuscript is difficult to read with many sentences inconsistent with the styles of native English speaker or even showing grammar errors. The manuscript must be refined by professional English writing services before publication.

Secondly, in introduction section, please find and review related literatures that descript similar systems and summarize the contributions of this paper.

Thirdly, in hardware development section, please add information on three aspects. 1). specifics of the UAV regarding its durance, payload, etc.,  2). specifics of the camera like the weight, sensor, etc., 3). communication system used to control the UAV and transfer the data between UAV and ground station.

Fourthly, in the result section, the author only provides the figures and tables without extracting the essential information from them to the reader. Please provide description of those figures and tables.

Finally, the discussion section is superficial and too short.

Author Response

Thank you for your valuable feedback. 

Based on the comments we have made, we have revised and rectified them. 

The modifications are uploaded with a file

Appreciate your valuable comments again,

Deokjin Lee

Reviewer 2 Report

The article presents a concept of airborne objects tracking system. It consists of a drone, commercial high computing power platforms and modified open-source software systems for tracking objects. Advantages of the article: 1) A concept of airborne object tracking system is presented. 2) The system was implemented and validated. 3) Initial assessment of performance of the system in various configurations is presented. Disadvantages of the article: 1) Most parts of the system, hardware and software, are outsourced. 2) Assessment of the object tracking is purely qualitative. 3) Energy consumption per tracked object has not been even revealed. It has not been compared for various versions of the system under test. 4) The system has not been compared to other systems known from literature. 5) There are some editing defects. For example acronym ADAS is used twice but not explained.

p { margin-bottom: 0.25cm; line-height: 115%; }

Author Response

(The authors gave the same response as above.)

Round 2

Reviewer 1 Report

Thanks to the author for responding to my comments. While the author addressed most of my comments appropriately, the language still does not live up to the publication standard. Refining the language by professionals is strongly suggested. 

Author Response

We've modified our manuscript by following the comments from the reviewers.

Specially, all the grammatical errors are checked and sentences are corrected by

reviewing again.

English was modified through the English revision center.

We hope that everything will be ok for the possible publication.

Thanks

Deokjin Lee

Reviewer 2 Report

The authors implemented my advises in the text. I am pleased. Thank you.

Author Response

(The authors gave the same response as above.)
